# Positive correlation between transcriptomic stemness and PI3K/AKT/mTOR signaling scores in breast cancer, and a counterintuitive relationship with *PIK3CA* genotype

Ralitsa R. Madsen[1]*, Emily C. Erickson[2], Oscar M. Rueda[3,4,5], Xavier Robin[6], Carlos Caldas[3,4,5], Alex Toker[2], Robert K. Semple[7●], Bart Vanhaesebroeck[1●]

1 University College London Cancer Institute, Paul O'Gorman Building, University College London, London, United Kingdom, 2 Department of Pathology, Medicine and Cancer Center, Beth Israel Deaconess Medical Center, Harvard Medical School, Boston, Massachusetts, United States of America, 3 Cancer Research UK Cambridge Institute and Department of Oncology, Li Ka Shing Centre, University of Cambridge, Cambridge, United Kingdom, 4 Cambridge Breast Unit, Addenbrooke's Hospital, Cambridge University Hospital NHS Foundation Trust, Cambridge, United Kingdom, 5 NIHR Cambridge Biomedical Research Centre and Cambridge Experimental Cancer Medicine Centre, Cambridge University Hospital NHS Foundation Trust, Cambridge, United Kingdom, 6 SIB Swiss Institute of Bioinformatics, Biozentrum, University of Basel, Basel, Switzerland, 7 Centre for Cardiovascular Science, Queen's Medical Research Institute, University of Edinburgh, Edinburgh, United Kingdom

● These authors contributed equally to this work.
* r.madsen@ucl.ac.uk

**Data Availability Statement:** All annotated analyses code/files are deposited on the Open Science Framework, alongside a 'how-to-usage-

## Abstract

A PI3Kα-selective inhibitor has recently been approved for use in breast tumors harboring mutations in *PIK3CA*, the gene encoding p110α. Preclinical studies have suggested that the PI3K/AKT/mTOR signaling pathway influences stemness, a dedifferentiation-related cellular phenotype associated with aggressive cancer. However, to date, no direct evidence for such a correlation has been demonstrated in human tumors. In two independent human breast cancer cohorts, encompassing nearly 3,000 tumor samples, transcriptional footprint-based analysis uncovered a positive linear association between transcriptionally-inferred PI3K/AKT/mTOR signaling scores and stemness scores. Unexpectedly, stratification of tumors according to *PIK3CA* genotype revealed a "biphasic" relationship of mutant *PIK3CA* allele dosage with these scores. Relative to tumor samples without *PIK3CA* mutations, the presence of a single copy of a hotspot *PIK3CA* variant was associated with lower PI3K/AKT/mTOR signaling and stemness scores, whereas the presence of multiple copies of *PIK3CA* hotspot mutations correlated with higher PI3K/AKT/mTOR signaling and stemness scores. This observation was recapitulated in a human cell model of heterozygous and homozygous *PIK3CA^{H1047R}* expression. Collectively, our analysis (1) provides evidence for a signaling strength-dependent PI3K-stemness relationship in human breast cancer; (2) supports evaluation of the potential benefit of patient stratification based on a combination of conventional PI3K pathway genetic information with transcriptomic indices of PI3K signaling activation.

guide", via the following link: https://osf.io/g8rf3/wiki/home/ & doi: 10.17605/OSF.IO/G8RF3 Raw and processed RNA sequencing data are deposited with GEO under accession number: GSE184277.

**Funding:** R.R.M. is supported by a Sir Henry Wellcome Fellowship (220464/Z/20/Z). R.K.S. is supported by the Wellcome Trust (210752/Z/18/Z). Work in the laboratory of B.V. is supported by Cancer Research UK (C23338, A25722) and PTEN Research. O.M.R. and C.C. are supported by Cancer Research UK. This work was supported by grants from the NCI (R35 CA253097 to A.T. and F31CA254000 to E.C.E.). The funders had no role in study design, data collection and analysis, decision to publish, or preparation of the manuscript.

**Competing interests:** I have read the journal's policy and the authors of this manuscript have the following competing interests: R.K.S. consults for Novartis (Cambridge, MA, USA) on use of PI3K inhibitors in overgrowth syndromes and has consulted for HotSpot Therapeutics (Boston, MA, USA); B.V. is a consultant for iOnctura (Geneva, Switzerland), Venthera (Palo Alto, CA, USA), Olema Pharmaceuticals (San Francisco, US) and Pharming (Leiden, The Netherlands).

## Author summary

Breast cancers often have increased activity of the so-called PI3Kα enzyme and the pathway it activates, usually attributed to genetic alterations in the *PIK3CA* gene, encoding a critical PI3Kα component. Recent cell studies have shown that effects of a *PIK3CA* mutation depend on how many copies are present. For example, two copies of a strong mutation, but not one, fix cells in a state of "stemness", a property associated with tumor aggressiveness and therapy failure. To determine relationships among PI3K genetic variation, PI3K activity and stemness in breast cancers we used data from independent patient cohorts encompassing nearly 3,000 tumors. Using PI3K signaling or stemness scores derived from gene expression data, we found a strong, positive association between the scores: aggressive tumors show the highest scores. In contrast, the relationship of these scores with *PIK3CA* mutation status was unexpected–cancers with one *PIK3CA* mutant copy showed a *decrease* in both scores, while they *increased* in cancers with additional copies. This was confirmed in cellular models. This suggests that using binary information about a *PIK3CA* mutation to define patient groups for trials may miss important effects of allele dosage. We suggest that grouping may be improved by combining *PIK3CA* mutational information with functional indices of PI3K pathway activation.

## Introduction

Activating mutations in *PIK3CA* are among the most common somatic point mutations in cancer, together with inactivation or loss of the tumor suppressor PTEN, a negative regulator of class I phosphoinositide 3-kinase (PI3K) enzymes [1–3]. PI3Kα-selective inhibitors are now making good progress in the clinic [4], with the PI3Kα-specific inhibitor alpelisib (Piqray/NVP-BYL719; Novartis) receiving approval for the treatment of advanced hormone-receptor (HR)-positive, HER2-negative breast cancers, in combination with the estrogen receptor (ER) antagonist fulvestrant [5]. The randomized phase III trial concluded that a clinically-relevant benefit of the combination therapy was more likely in patients with *PIK3CA*-mutant tumors [5]. The FDA approval of alpelisib was accompanied by approval of the companion diagnostic therascreen *PIK3CA* test (QIAGEN) which detects 11 *PIK3CA* hotspot mutations. Despite these advances, a substantial proportion of patients with *PIK3CA*-mutant tumors failed to improve on the combination therapy [5], highlighting the need for further refinement of current patient stratification strategies.

Experimental evidence suggests that heterozygous expression of a strongly activating *PIK3CA* mutation alone is insufficient to transform cells *in vitro* or to induce tumorigenesis *in vivo* (reviewed in Ref. [6]). This is supported by observations of people with disorders in the *PIK3CA*-related overgrowth spectrum (PROS) which is caused by the same *PIK3CA* mutations found in cancer, but does not feature discernible excess risk of adult malignancy [6]. It thus appears that additional events are required for cell transformation, possibly in the PI3K pathway itself. In this regard, we and others have shown that many *PIK3CA*-associated cancers harbour multiple independent mutations activating the PI3K pathway, including multiple *PIK3CA* mutations in *cis* or *trans* [3,7–10].

Overexpression of wild-type *Pik3ca* or the hotspot *Pik3ca^{H1047R}* mutation has been linked to dedifferentiation and stemness in murine models of cancer [11–17], particularly of the breast, but *Pik3ca* gene dose-dependent regulation was not addressed in these studies. Pluripotent stem cells (PSCs) share key characteristics with cancer cells, including developmental plasticity, the capacity for indefinite self-renewal, rapid proliferation and high glycolytic flux [18].

We recently reported that human PSCs with two endogenous alleles of the strongly activating cancer hotspot mutation *PIK3CA*$^{H1047R}$ exhibit pronounced phenotypic differences compared to isogenic cells heterozygous for the same *PIK3CA* variant [8]. These differences include partial loss of epithelial morphology, widespread transcriptional reprogramming and self-sustained stemness *in vitro* and *in vivo*, none of which were observed in heterozygous *PIK3CA*$^{H1047R}$ cells [8]. Collectively these findings emphasize the importance of *PIK3CA* mutation dose, and its inferred functional correlate, PI3K signaling strength, in determining the cellular consequences of mutational activation of this pathway. In line with this notion, dose-dependent effects on stemness have also been reported in mouse embryonic stem cells with heterozygous *versus* homozygous loss of *Pten* [19].

Stemness or dedifferentiation, accompanied by re-expression of embryonic genes, is a feature of aggressive tumors [20,21]. Beyond direct histopathological analyses, this has been supported by computational analyses examining a tumor's expression of defined PSC gene signatures [20–23]. With the continuing collection and curation of multi-omics datasets by the cancer community, such signatures can now be employed *en masse* to study how cancer-specific stemness relates to other biological processes of interest. This can, however, be challenging for highly dynamic processes such as signaling pathway activity which is best inferred using temporal protein-based measurements. Such measurements are not available for most human tissue samples. A complementary approach is the use of transcriptional "footprints" of pathway activation, derived from the systematic curation of gene expression data obtained from direct perturbation experiments [24–26]. Given the slower time scale of gene expression regulation relative to acute signaling changes at the protein level, transcriptional footprint analyses can be thought of as an integrated measure of pathway activity over a longer time scale. The power of such analyses has been best demonstrated by The Connectivity Map Resource, which enables discoveries of gene and drug mechanisms of action on the basis of common gene-expression signatures [27,28].

Here, we set out to determine whether a signaling strength-dependent PI3K-stemness link exists in human breast cancer, and to provide a systematic characterization of relevant clinical and biological correlates. We used established, open-source methods to infer PI3K/AKT/mTOR (henceforth PI3K signaling) and stemness scores from publicly available transcriptomic data from nearly 3,000 primary human breast tumors. Our analyses reveal a positive, linear relationship between PI3K signaling and stemness scores, and uncover a surprising "biphasic" relationship between these scores and mutant *PIK3CA* allele dosage. This suggests a potential utility for combined functional genomics and genotype assessments in future patient stratification for PI3K-targeted therapy. Consistent with prior cell biology studies, breast tumor transcriptomic analyses revealed strong clustering of PI3K/AKT/mTOR and stemness scores with MYC-related biological processes, including proliferation and glycolysis. With the advent of routine tumor gene expression analyses, further dissection of the mechanisms driving these associations may enable much-needed further therapeutic advances.

## Results

### Transcriptional indices of PI3K signaling activity in breast cancer are positively associated with stemness and tumor grade

The molecular features of stemness can be captured by gene signatures derived by computational comparisons of pluripotent stem cells and differentiated derivatives. Among the first such signatures was PluriNet (n = 299 genes; **S1 Table**), generated with machine learning methods [29], and applied below to primary breast cancer samples. To evaluate PI3K pathway

activity in the same samples, we used the "HALLMARK_PI3K_AKT_MTOR_SIGNALING" gene signature from the Broad Institute Molecular Signature Database (MSigDB). This gene signature consists of 105 genes upregulated upon PI3K pathway activation across multiple studies [25] (**S2 Table**), thus corresponding to a gene expression footprint of PI3K pathway activation. Of note, only 4 genes were shared between the PI3K/AKT/mTOR and PluriNet gene lists, precluding a direct confounding effect on the relationship between stemness and PI3K signaling scores tested here.

We used Gene Set Variation Analysis (GSVA) [30], an open-source enrichment analysis method, to calculate stemness and PI3K signaling scores on the basis of these gene expression signatures, independently in breast cancer tumors with available transcriptomic data from the METABRIC ($n$ = 1980; used for primary analyses) and TCGA patient cohorts (n = 928; used for secondary analyses). Similar to the conventional gene set enrichment analysis (GSEA), GSVA is a non-parametric method that evaluates the concerted behavior of functionally related genes; in contrast to GSEA, however, the GSVA method is unsupervised and enables functional enrichment analyses beyond conventional case-control experimental designs [30]. This makes the GSVA method ideally suited for pathway-centric analyses of transcriptomic data with diverse clinical and phenotypic correlates, conditional upon a relatively large sample size (n > 30) [30].

Consistently, the GSVA-derived PI3K signaling score in METABRIC breast tumors correlated significantly with the stemness score (**Fig 1A**; Spearman's Rho = 0.5, p < 2.2e-16) as well as tumor grade status (**Fig 1B**), a measure of tumor dedifferentiation based on histopathological assessment. A similar linear relationship between PI3K signaling and stemness scores was also found in TCGA breast cancers (**Fig 1C**; Spearman's Rho = 0.4; p < 2.2e-16).

To ascertain the ability of this approach to capture *bona fide* features of stemness and PI3K signaling from transcriptomic data, we next performed pairwise-correlations with independently-derived transcriptomic indices for each phenotype. Across both METABRIC (**Fig 1D**) and TCGA (**Fig 1E**) breast tumors, the PluriNet-derived stemness score showed good concordance with alternative stemness scores obtained using the one-class logistic regression (OCLR)-based signature of Malta *et al.* [22], or the signature from Miranda *et al.* [23], a modified version of a gene set initially developed by Palmer *et al.* [21]. The strongest correlations (Spearman's Rho > 0.7) were between PluriNet and the OCLR-based signature, both of which were derived using distinct machine learning algorithms.

Next, we evaluated the concordance of our PI3K signaling score with complementary measures of pathway activity. First, we applied PROGENy to obtain an independent measure of PI3K pathway activation based on transcriptomic footprints. Instead of the enrichment score calculated by GSVA, PROGENy uses a linear model to infer pathway activity from the expression of 100 pathway-responsive genes [24]. The GSVA- and PROGENy-derived PI3K signaling scores exhibited a significant positive correlation across both METABRIC (Spearman's Rho = 0.61) and TCGA (Spearman's Rho = 0.45) breast cancers (**Fig 1D** and **1E**). Similar positive associations were obtained when we used GSVA to calculate a PI3K signaling score based on two alternative PI3K-response gene signatures ("PI3K_Jin_1" and "PI3K_Jin_2"), which were recently shown to associate positively with human breast cancer metastases in the brain [31]. All these PI3K signaling scores exhibited consistently significant, positive correlations with the stemness scores tested above (**Fig 1D** and **1E**).

Finally, we sought independent experimental evidence that the "HALLMARK_PI3-K_AKT_MTOR_SIGNALING" gene signature can be used to identify biologically meaningful activation/inhibition of the PI3K pathway, as suggested by the methodology used to generate this transcriptional footprint [25]. To this end, we used untransformed, immortalized human breast epithelial MCF10A cells with stable *PIK3CA^H1047R* overexpression, alongside the

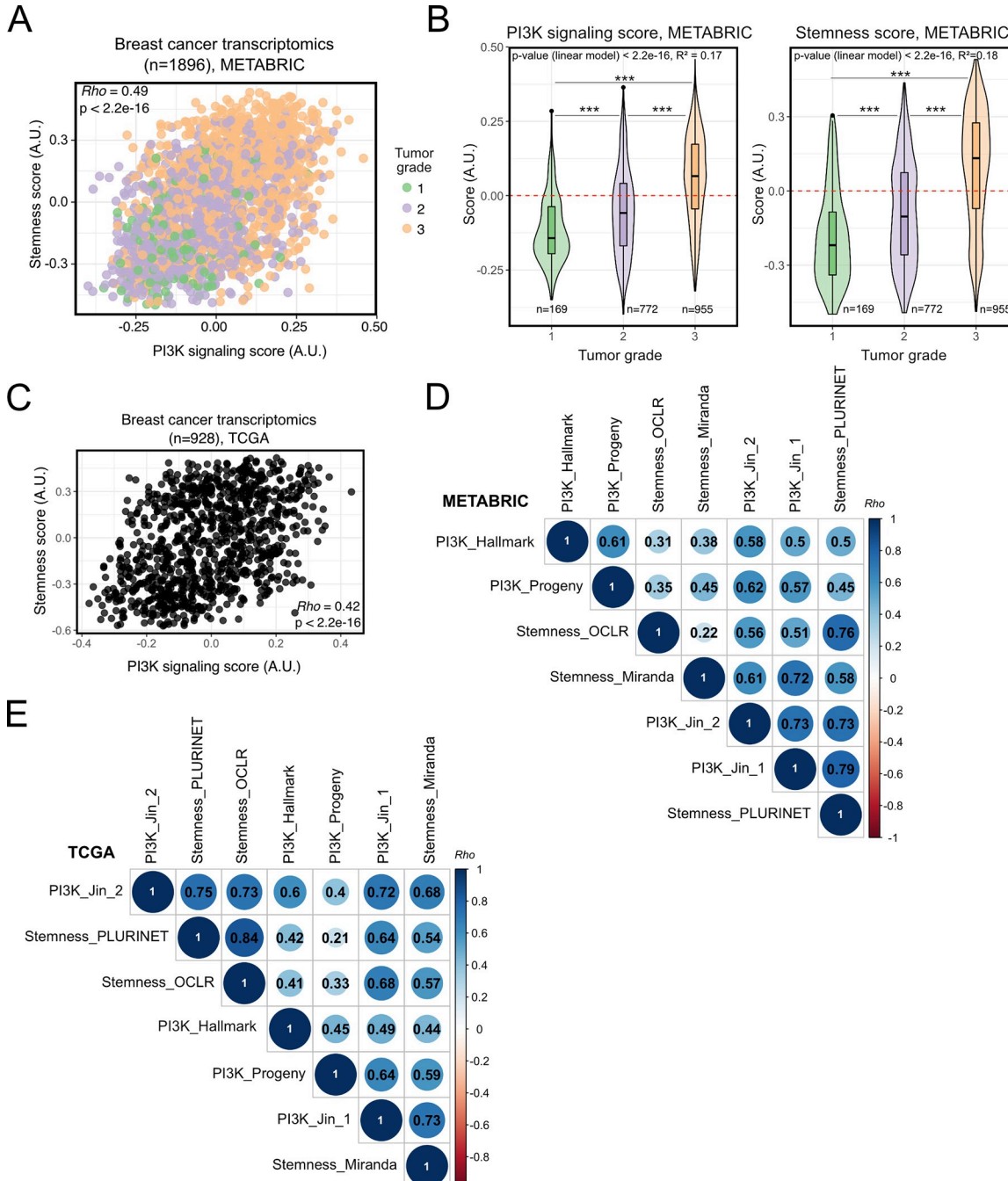

**Fig 1. Transcriptionally-inferred PI3K pathway activation and breast tumor stemness/grade exhibit significant, positive association. (A)** Rank-based (Spearman's *Rho*) correlation analysis of the relationship between transcriptionally-inferred PI3K pathway activity and stemness scores, evaluated across METABRIC breast cancer transcriptomes. Scores were determined using Gene Set Variation Analysis (GSVA) with mSigDB "HALLMARK_PI3K_AKT_MTOR_SIGNALING" (for PI3K activity score) and "MUELLER_PLURINET" (for stemness score) gene signatures [25,29,30]. Gene lists used are included in Supplementary Tables 1 and 2. **(B)** PI3K signaling and stemness score distributions across breast cancer grade (METABRIC). *** p ≤ 0.001 according to one-way ANOVA with Tukey's Honest Significant Differences method. The global p-value for each linear model is indicated within each plot. **(C)** As in (A) but based on TCGA breast invasive carcinoma (BRCA) transcriptomic data. **(D)** Rank-based correlation analyses of the stemness (PluriNet-based) and PI3K (mSigDB-based) scores used in the current study against published and independently-derived transcriptional indices for stemness and PI3K signaling, across METABRIC breast cancer transcriptomic data. Individual *Rho* coefficients are shown within the respective circles whose sizes are matched accordingly. Only significant correlations are shown (family-wise error rate < 0.05). **E)** As in (D) but based on TCGA BRCA

transcriptomic data. The Stemness_OCLR score is based on a machine-learning-derived stemness signature [22]; the Stemness_Miranda score is based on a modification of the stemness signature of Palmer et al. [21,23]. The PI3K_Progeny score is based on the analysis of benchmarked pathway-responsive genes as described in Ref. [24]. The "PI3K_Jin_1" and "PI3K_Jin_2" gene signatures were obtained from Ref. [31]. Note that the scores calculated using OCLR and PROGENy are independent of the GSVA method used to generate scores based on the remaining signatures.

respective empty vector (EV) control cells [32]. Assessed in growth factor-replete conditions to reflect a more physiological environment, $PIK3CA^{H1047R}$ cells exhibited the expected increase in p110α expression and PI3K/AKT/mTOR1 pathway activation as measured by phosphorylation of AKT on S473 (pAKT) and T246 on PRAS40 (pPRAS40) (**Fig 2A**). Treatment of $PIK3CA^{H1047R}$ MCF10A cells with 500 nM of the p110α-selective inhibitor BYL719 (alpelisib/Piqray; Novartis) restored pAKT and pPRAS40 levels to those in control cells, both after 48 h and 120 h of drug exposure (**Fig 2A**). Moreover S240/S244 phosphorylation of S6 (pS6)–a marker of mTORC1 activity–in $PIK3CA^{H1047R}$ MCF10As cells treated with BYL719 decreased below those observed in EV control cells (**Fig 2A**). We next performed bulk mRNA sequencing of replicate samples in the same conditions, asking if the results from a conventional GSEA with the "HALLMARK_PI3K_AKT_MTOR_SIGNALING" gene set would reflect the observed biochemical changes. The samples exhibited low intra-group variability and clustered according to $PIK3CA^{H1047R}$ expression and BYL719 treatment based on unsupervised, principal component analysis (PCA) (**Fig 2B**). Given possible transcriptional delays, we analyzed both time points to ensure that we would capture the response to p110α inhibition. While the observed enrichments for the "HALLMARK_PI3K_AKT_MTOR_SIGNALING" gene set were modest (**Fig 2C**), as expected from the biochemical signaling data, the direction of change across the different comparisons was consistent with our hypothesis–positive when evaluating MCF10A cells with $PIK3CA^{H1047R}$ overexpression relative to the corresponding EV controls, and negative upon treatment of $PIK3CA^{H1047R}$-overexpressing cells with BYL719 (**Fig 2C**). Similar results were obtained with the related "HALLMARK_MTORC1_SIGNALING" gene set (**Fig 2C**), however upon further testing we found that the relative shifts for this signature were not robust to differences in background gene expression (**S1 Fig**). We note that the relative shifts in calculated scores are the basis for all our subsequent breast cancer analyses in the context of GSVA-based score calculations, as per published examples [30].

Taken together, these results provide evidence that: 1) pre-defined transcriptional footprints for the PI3K pathway can be used to obtain a biologically meaningful score for pathway activity from transcriptomic data; 2) there is a positive, "strength"-dependent relationship between stemness and overall PI3K signaling in human breast cancer.

## Stemness and PI3K signaling scores differ across breast cancer subtypes

Using our GSVA-based stemness and PI3K signaling scores, we next sought to determine their relationship with clinical breast cancer subtype. Upon stratification of METABRIC breast cancers into those with "high" and "low" PI3K signaling scores, we found that around 45% of tumors with a high PI3K signaling score were ER-negative, in contrast to 4% of tumors with low PI3K signaling scores (**Fig 3A**). In TCGA, the corresponding percentages were 33% and 7% (**S2A Fig**). Consistently, PI3K signaling and stemness scores were highest in the more aggressive PAM50 breast cancer subtypes (**Fig 3B**), including Basal, HER2 and Luminal B. These findings are in line with independent studies relying on alternative indices and methods for quantifying PI3K signaling and stemness in separate analyses [20,22,31,33–35]. Importantly, the correlation of a high PI3K signaling score with ER-negativity contrasts with the

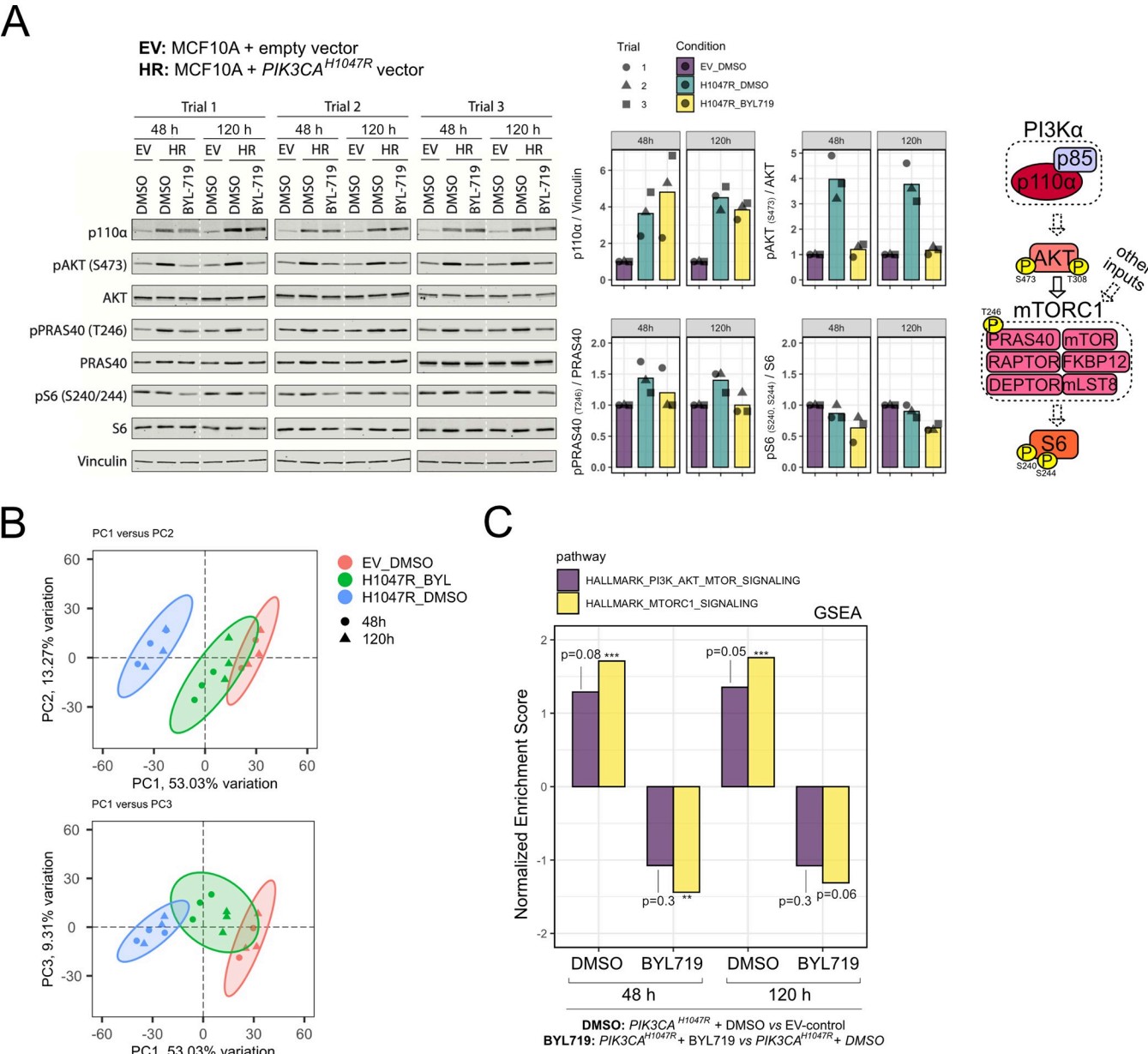

**Fig 2. Experimental validation of the PI3K signaling score in a breast epithelial cell model with oncogenic PI3Kα activation (*PIK3CA^H1047R*). (A)**
Western blotting for canonical class IA PI3K pathway signaling components in MCF10A cells with retroviral overexpression of *PIK3CA^H1047R* -/+ 500 nM BYL719, compared to empty vector (EV) control cells. Samples were collected after 48 h and 120 h of inhibitor treatment in growth factor-replete medium. All samples from the same experimental trial were loaded on the same gel, with the stippled white line included to emphasize the different time points. The quantified data are shown as barplots with the corresponding replicate points. All targets were normalized to a corresponding total protein as indicated on the y axis, in addition to normalization to the EV_DMSO condition within each experimental trial and time point. The diagram on the right-hand side represents a simplified schematic of PI3Kα signaling, with stippled arrows for indirect regulation; known negative feedback loops have been omitted for clarity. **(B)** Principal component (PC) analysis of transcriptomic data corresponding to samples in (A) except for exclusion of a single EV_DMSO_48 h replicate due to technical issues with the mRNA library. **(C)** Gene set enrichment analysis (GSEA) following differential gene expression analysis and gene ranking with respect to the indicated comparisons (*PIK3CA^H1047R* + DMSO *versus* EV-control; *PIK3CA^H1047R* + BYL719 *versus* EV-control) at 48 and 120 h. The analyses were performed with the "HALLMARK_PI3K_AKT_MTOR_SIGNALING" (105 genes; 90 present in the ranked gene list) and the "HALLMARK_MTORC1_SIGNALING" (200 genes; 192 present in the ranked gene list) gene sets as indicated (number of overlapping genes between the two gene sets = 24). The total number of ranked genes was 11,777. The p-values correspond to each enrichment's significance following 100,000 permutations of the gene ranks; * $p \leq 0.05$, ** $p \leq 0.01$, *** $p \leq 0.001$; FDR = 0.05. See also S1 Fig.

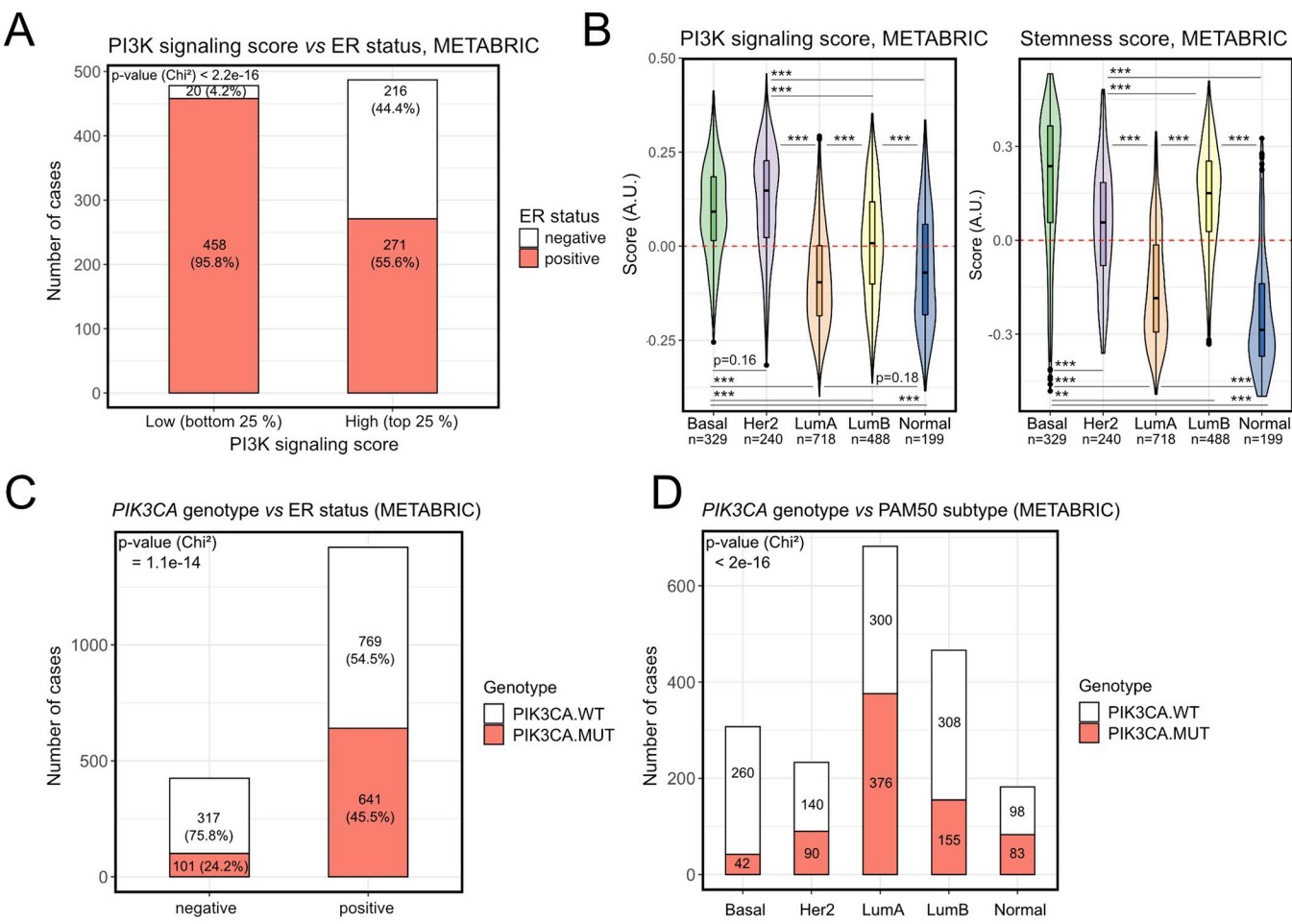

**Fig 3. High PI3K signaling and stemness scores, but not *PIK3CA* mutations, are enriched in aggressive breast cancer subtypes. (A)** PI3K signaling score distribution in METABRIC breast tumors stratified according to ER status. **(B)** PI3K signaling and stemness score distributions across METABRIC breast cancers stratified according to PAM50 subtype; ** p ≤ 0.01, *** p ≤ 0.001 according to Tukey's Honest Significant Differences method. **(C)** and **(D)** The distribution of *PIK3CA* wild-type (PIK3CA.WT) and mutant (PIK3CA.MUT) samples in METABRIC breast cancers, stratified according to ER status (C) or PAM50 subtype (D).

known enrichment of *PIK3CA* mutations in ER-positive breast tumors [35,36], which were also reproduced by our analyses (**Fig 3C and 3D**).

## PI3K signaling and stemness scores, but not binary *PIK3CA* mutant status, predict prognosis in breast cancer

As expected, given the positive association between PI3K signaling and stemness scores with tumor grade, both scores were negatively associated with patient survival in the METABRIC cohort, with a clear dosage relationship between the assessed scores and survival, including progressively worsened survival in tumors with high *vs* intermediate *vs* low scores (**Fig 4A and 4B**). This relationship was not simply driven by the above-mentioned enrichment of high PI3K signaling and stemness scores in more aggressive ER-negative tumors, as the prognostic power of both scores remained when evaluated in ER-positive tumors only (**Fig 4C and 4D**). In contrast, although overall ER-negative cases with available survival data were limited in number, we in fact noticed a loss of prognostic power when evaluating the two scores in this breast cancer subset (**S2B and S2C Fig**). Due to limited data, extensive survival analyses were

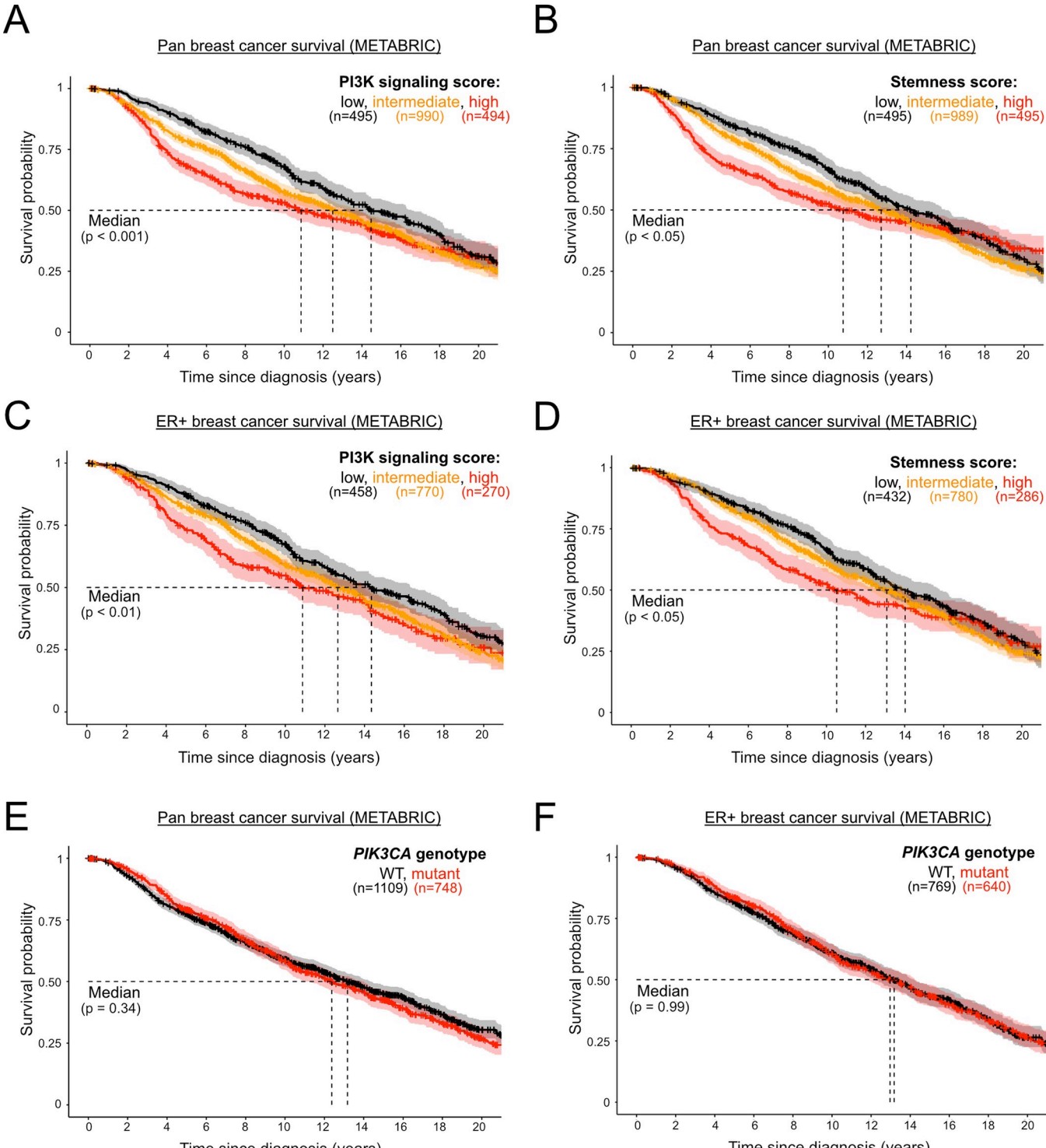

**Fig 4. PI3K signaling and stemness scores, but not *PIK3CA* genotype, are prognostic in ER+ breast cancer.** Pan-breast cancer patient survival in METABRIC, as a function of PI3K signaling (**A**) or stemness (**B**) score. Survival analysis in estrogen receptor (ER)-positive breast cancer patients, as a function of PI3K signaling (**C**) or stemness (**D**) score. Low, intermediate and high classifications represent the bottom quartile, the interquartile range and the top quartile of the respective scores. (**E**) and (**D**) represent pan- and ER-positive breast cancer patient (METABRIC) survival, respectively, as a function of binary *PIK3CA* genotype. The mutant genotype captures only cases with activating missense mutations. The sample size for each panel and subgroup is indicated, and p-values were calculated using a log-rank test. The 95% confidence intervals are indicated by shading.

not possible in TCGA breast cancers, however the negative association between PI3K signaling "strength" and pan-breast cancer survival was reproduced (**S2D Fig**).

As previously reported [36–38], *PIK3CA* mutant status (hotspot and non-hotspot) had no prognostic power in pan-breast or ER-positive METABRIC tumors, despite their enrichment in the ER-positive cohort (**Fig 4E** and **4F**). Interestingly, however, the presence of *PIK3CA* mutations in ER-negative tumors appeared to be associated with worse prognosis (**S2E Fig**).

## Stratification of breast cancers by mutant *PIK3CA* allele dosage reveals a biphasic relationship with PI3K signaling and stemness scores

Given the divergent correlations between PI3K signaling scores and *PIK3CA* mutant status in the survival analyses, we next assessed the relationship between stemness/PI3K signaling scores and *PIK3CA* genotype, taking into account available information on mutant *PIK3CA* allele dosage on the basis of our previous work with TCGA tumors [8]. For METABRIC, we inferred *PIK3CA* copy number changes based on available information on allele gain/amplification in cBioPortal. For both cohorts, we specifically focused on tumors harboring one or more hotspot *PIK3CA* alleles, given the well-established increased cellular activity of these mutants and their association with disease severity [39–42].

As PI3K pathway activation and tumor dedifferentiation can be triggered by a range of oncogenic hits, the relatively high PI3K signaling and stemness scores in breast cancers with wild-type *PIK3CA* were not entirely surprising (**Fig 5A** and **5B**). It was, however, counterintuitive that the presence of a single oncogenic *PIK3CA* missense variant was associated with a substantial reduction in the stemness score and a modest reduction in the PI3K score (**Fig 5A** and **5B**). Relative to tumors with a single *PIK3CA* mutant copy, those with multiple oncogenic *PIK3CA* copies exhibited higher PI3K signaling and stemness scores (**Fig 5A** and **5B**). This relationship was lost upon simple binary classification based on *PIK3CA* genotypes (i.e. wild-type *vs* mutant) (**Fig 5A** and **5B**). The observed biphasic relationship also remained upon stratification of tumors according to genome doubling (data only available for TCGA samples; **Fig 5C**).

Surprised by this observation, we next asked whether the biphasic relationship between *PIK3CA* genotype and transcriptionally-derived PI3K signaling/stemness scores could be recapitulated in a controlled cellular model. We turned to human induced pluripotent stem cells (iPSCs) that we engineered previously to harbor heterozygous or homozygous *PIK3CA*$^{H1047R}$ alleles, the only reported cellular models of heterozygous and homozygous *PIK3CA*$^{H1047R}$ expression on an isogenic background to date [43]. Using the published high-depth transcriptomic data on *PIK3CA*$^{WT/H1047R}$ and *PIK3CA*$^{H1047R/H1047R}$ iPSCs [43], we next performed GSEA with the two gene set signatures used for PI3K signaling and stemness score calculations in the breast cancer setting (MSigDB "HALLMARK_PI3K_AKT_MTOR_SIGNALING" and PluriNet, respectively). In line with their established biochemical and cellular phenotypes [8,43], homozygous *PIK3CA*$^{H1047R}$ iPSCs showed strong positive enrichment for both PI3K/AKT/mTOR and stemness gene signatures (**Fig 5D**). In contrast, their heterozygous *PIK3CA*$^{H1047R}$ counterparts presented with a strong negative enrichment for stemness, and a negative albeit statistically insignificant enrichment for the transcriptional PI3K/AKT/mTOR signature (**Fig 5D**). These patterns mirror those observed in human breast cancers and corroborate the existence of a previously unappreciated biphasic relationship between *PIK3CA* allele dosage and stemness.

## Breast cancer PI3K signaling and stemness scores are positively associated with proliferative and metabolic processes

Given the high depth and large sample size of the available breast cancer transcriptomic data, we next undertook a global analysis encompassing all 50 "hallmark" MSigDB gene sets and the

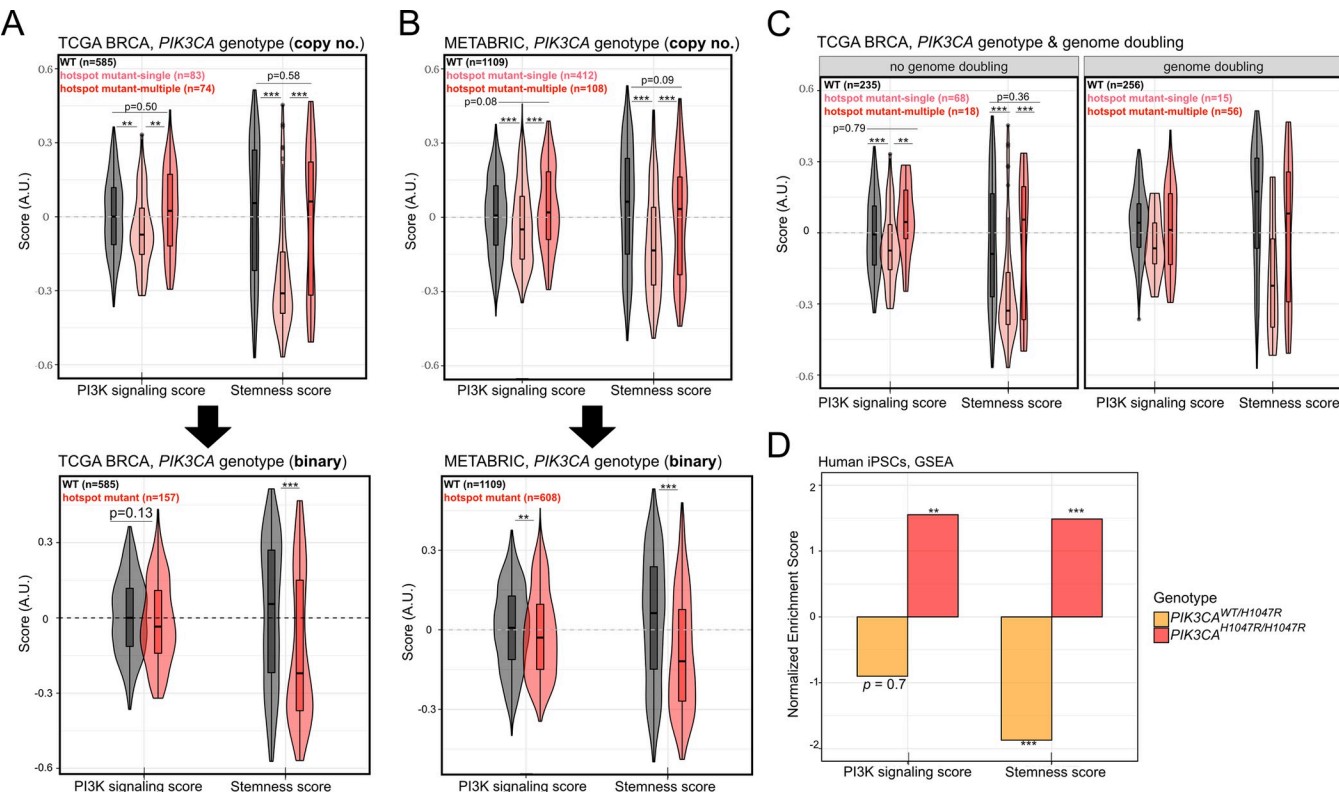

**Fig 5. The presence of a single-copy, but not multi-copy, hotspot *PIK3CA* mutation is associated with lower PI3K signaling and stemness score. (A)** PI3K signaling and stemness score distributions across TCGA breast cancers following stratification according to the presence or absence of single *vs* multiple copies of *PIK3CA* "hotspot" variants (C420R, E542K, E545K, H1047L, H1047R); ** p ≤ 0.01, *** p ≤ 0.001 according to one-way ANOVA with Tukey's Honest Significant Differences method. **(B)** As in (A) but performed using METABRIC breast cancer transcriptomic and genomic data. **(C)** As in (A) but further stratified according to available genome doubling information. The shown statistics apply to both subpanels and are the result of a two-way ANOVA, followed by Tukey's Honest Significant Differences method to determine differences between the indicated *PIK3CA* genotypes following adjustment for genome doubling. **(D)** Complementary GSEA-based PI3K signaling and stemness score calculations using publicly-available transcriptomic data from iPSCs with heterozygous or homozygous *PIK3CA^{H1047R}* expression [43]; enrichments are calculated relative to isogenic wild-type controls. ** p ≤ 0.01, *** p ≤ 0.001 for individual enrichments, according to FDR = 0.05 (Benjamini-Hochberg correction for multiple comparisons).

PluriNet signature to identify relevant biological processes associated with breast cancer stemness and a high PI3K signaling score. Such processes can be used to guide future experimental studies aimed at dissecting the molecular underpinnings of the observed relationships. To identify such associations, we applied GSVA to METABRIC and TCGA data to generate a score for each gene signature, followed by correlation analysis with hierarchical clustering. This global approach also allowed us to confirm that we are able to identify biologically-relevant gene signature clusters more broadly. For example, gene signatures associated with inflammatory processes clustered together according to strong pairwise positive correlations in both METABRIC and TCGA datasets (**Fig 6 bottom cluster, Fig 7 top left cluster**).

Data from either breast cancer cohort revealed a characteristic clustering pattern for PI3K signaling and stemness scores, including strong positive associations with proliferative (e.g., "G2M_checkpoint", "E2F_targets", "MYC_targets") and metabolic (e.g., "Glycolysis", "Oxidative_phosphorylation", "Reactive_oxygen_species") gene signatures (**Figs 6** and **7**). These signatures shared few genes (**S2F Fig**), ruling out technical artefacts as a source of the positive associations. Given prior observations of a strong correlation between PluriNet and cell cycle signatures, it has been suggested that PluriNet genes function as a distinct module within a larger context including cell cycle-specific genes [29]. Notably, the separate mTORC1 gene

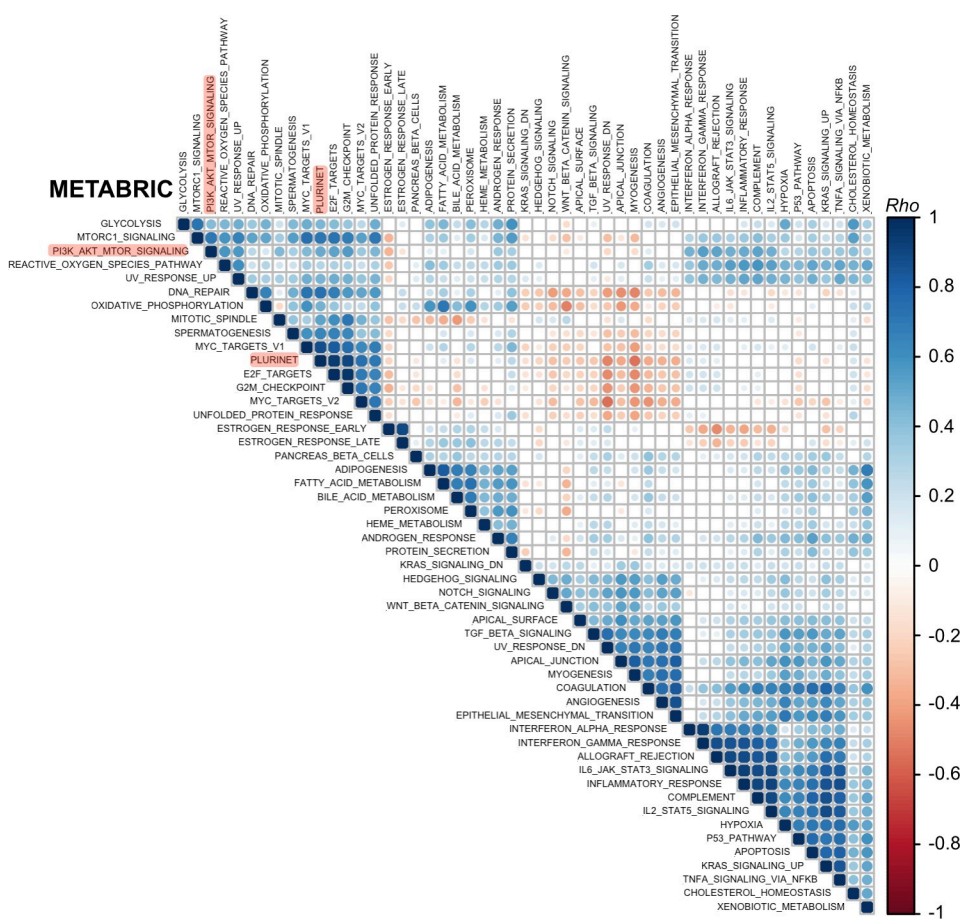

**Fig 6. Breast cancer (METABRIC cohort) PI3K signaling and stemness scores form a common cluster with proliferative and metabolic processes.** Rank-based correlation analyses across METABRIC GSVA-derived gene set enrichment scores, evaluating all 50 mSigDB Hallmark Gene Sets and PluriNet. Individual *Rho* coefficients are shown within the respective circles whose sizes are matched accordingly. Only significant correlations are shown (family-wise error rate < 0.05). The clusters were generated using unsupervised hierarchical clustering. The positions of PluriNet (stemness) and PI3K_AKT_MTOR (PI3K signaling) signatures are highlighted in red.

signature exhibited a much stronger correlation (Spearman's rho = 0.7) with the stemness score compared with the PI3K_AKT_mTOR signature used to derive the PI3K signaling score, on par with the correlation values observed between PluriNet and cell cycle signatures.

## Discussion

This study provides a comprehensive analysis of the relationship between PI3K signaling and stemness (or tumor dedifferentiation) using two large breast cancer transcriptomic datasets encompassing almost 3,000 primary tumors. We demonstrate a strong, positive relationship between transcriptionally-inferred PI3K signaling strength, stemness gene expression and histopathological tumor dedifferentiation. Importantly, we show that stratification of breast tumors according to single *vs* multiple copies of *PIK3CA* hotspot mutations results in distinct and near-opposite distributions with respect to PI3K signaling and stemness scores, an observation that was recapitulated in a controlled cell model system.

The PI3Kα-specific inhibitor alpelisib (Piqray/NVP-BYL719; Novartis) recently received approval for use in combination with the ER-antagonist fulvestrant in the treatment of ER-

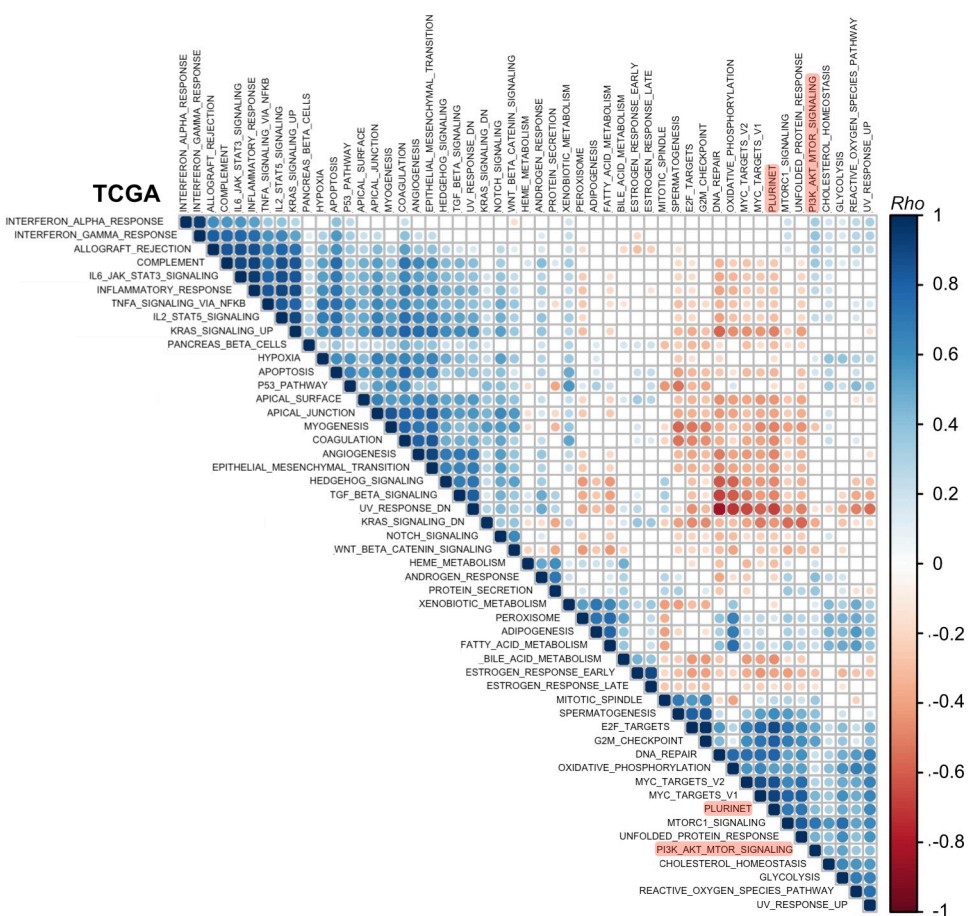

**Fig 7. Breast cancer (TCGA cohort) PI3K signaling and stemness scores form a common cluster with proliferative and metabolic processes.** As in Fig 6 except based on data from the TCGA breast cancer cohort as indicated.

positive breast cancers. The benefit of this treatment was most notable in *PIK3CA*-mutant tumors, yet the predictive value of binary mutant classification was incomplete [5]. This is a common observation for single gene biomarkers in cancer and has long spurred discussions about the utility of phenotypic pathway signatures for clinical response prediction [44]. It is therefore interesting to note that while *PIK3CA* mutations are enriched in the ER-positive breast cancer subgroup, on average these tumors also feature lower PI3K signaling and stemness scores as inferred from our transcriptional footprint analyses. The opposite is true for ER-negative tumors. Given that the MSigDB "HALLMARK_PI3K_AKT_mTOR_SIGNALING" signature used in our study also encompasses mTORC1-related processes, in line with a strong correlation with the separate hallmark mTORC1 signature, our findings support a previous study reporting a negative relationship between the presence of a *PIK3CA* mutation and mTORC1 signaling in ER-positive/HER2-negative breast cancers [38]. As we show, however, simple binary classification of tumors into *PIK3CA* wild-type and mutant genotypes, without allele dosage considerations, is likely to have masked a more complex relationship. Conversely, our study does not distinguish between AKT- and mTORC1-specific processes, which may nevertheless be important to consider for further mechanistic understanding and patient stratification [38,45,46]. Given the high correlation between the hallmark "PI3K_AKT_mTOR_-SIGNALING" and "mTORC1_SIGNALING" signature scores (Spearman's rho = 0.7), we

speculate that the observed relationship between PI3K and stemness in breast cancer may be driven by mTORC1-dependent processes.

Disentangling the apparent biphasic relationship between single *versus* multiple copies of *PIK3CA* mutation and stemness scores will require direct experimentation, but is likely to reflect context-dependent feedback loops within the intracellular signaling networks. Such feedback loops can result in non-intuitive and discontinuous outcomes upon different levels of activation of the same pathway, as demonstrated in our isogenic iPSC system with heterozygous and homozygous *PIK3CA^{H1047R}* expression [8,43]. In general, our observations caution against the use of a binary *PIK3CA*-mutant-centric approach to predict PI3K pathway activity outcomes. Moreover, we note that numerous alternative (epi)genetic changes–including *PIK3CA* amplification or increased mutant-specific mRNA expression, loss of *PTEN* or *INPP4B* –may converge on increased, and perhaps dose-dependent, PI3K pathway activation [3,33,35,47,48]. Importantly, such *PIK3CA* mutant-independent and/or non-genetic mechanism of PI3K pathway activation will be captured by the transcriptional footprint-based PI3K signaling scores used in our study and will thus contribute to the values observed in non-*PIK3CA* mutant tumors. More generally, the diagnostic and therapeutic benefits of combined, comprehensive genomic and non-genomic analyses were recently demonstrated in patients with rare cancers [49], and are worth considering in the context of breast and other cancers where PI3K pathway alterations feature prominently [3].

While PI3K signaling and stemness scores exhibited a strength-dependent negative association with patient survival pan-breast cancer as well as in ER-positive tumors, this prognostic power was not observed with binary genotype-based *PIK3CA* classification. Paradoxically, however, *PIK3CA* mutations had prognostic power in ER-negative tumors, in contrast to PI3K signaling and stemness scores. This raises the question whether subgroups defined by differences in *PIK3CA* mutant status and PI3K signaling/stemness scores differ in their response to PI3Kα-targeted therapy. PI3K/AKT inhibitors have so far had limited success in TNBC [50–53], the tumor subgroup with some of the highest PI3K signaling and stemness scores. In the context of our findings, it is interesting to note that studies in mouse models of metastatic breast cancer has demonstrated the potential utility of combining PI3K and BET inhibitors to overcome a MYC-dependent feedback mechanism that limits the benefit of single-agent PI3K pathway inhibition in this context [54].

It is also notable that our correlation analyses of breast cancer transcriptomes identified a PI3K/stemness cluster encompassing key processes associated with the MYC regulatory module in pluripotent stem cells [55]; a module previously shown to be active in various cancers and predictive of cancer outcome [56]. Moreover, computational analyses of iPSCs with homozygous *PIK3CA^{H1047R}* expression identified MYC as a central hub connecting the PI3K, TGFβ and pluripotency networks in these cells [43]. Recently, *PIK3CA^{H1047R}*/*KRAS^{G12V}* double knock-in breast epithelial cells were also shown to exhibit a high MYC transcriptional signature, when compared to single-mutant counterparts [57]. Collectively, the recurrent appearance of MYC in these independent analyses raises the possibility that this transcription factor governs the mechanistic link between stemness and PI3K signaling strength in pluripotent stem cells and breast cancer. Experimental studies will be required to test this hypothesis, alongside a potential involvement of mTORC1 as suggested by the observed strong positive correlation between the mTORC1 signature and stemness/MYC signatures.

A limitation of the current and previous bulk-tissue transcriptomic analyses is that they cannot determine (1) whether the observed correlations reflect mechanistic links or spurious associations caused by a confounder variable that influences two or more processes independently and (2) to what extent the observed transcriptomic scores are driven by changes in tumor subcellular composition, tumor cell type-specific phenotypic alterations, and/or non-

cell-autonomous interactions with the stroma. Nevertheless, given the ability to reproduce key observations in controlled cell model systems, our analyses of the relationship between PI3K signaling dose and stemness in breast cancer may prove useful in guiding future experimental studies aimed at identifying the exact molecular underpinnings. Since we know that heterozygous $PIK3CA^{H1047R}$ iPSCs exhibit moderate PI3K pathway activation at the biochemical level [8,43], the fact that this is not captured in a positive transcriptional footprint-based PI3K signaling score in this context is worth noting. Combined with the observation of an apparent decrease in the PI3K signaling score in tumors with a single copy of a hotspot *PIK3CA* mutation, we surmise that this may reflect feedback mechanisms that are sufficient to limit the influence of intermediate but not strong PI3K pathway activation in certain settings. This warrants further studies as it may have important consequences for targeting of tumors with a high *versus* low transcriptionally-inferred PI3K signaling score. It is also worth noting that previous protein-based signaling studies of breast cancer cell lines and tumors with and without *PIK3CA* mutations found that *PIK3CA* mutations were associated with lower and/or inconsistent PI3K pathway activation [33,36,38]. This also emphasizes the need for future benchmark studies that establish the relationship between the absolute magnitude of individual transcriptomic scores for the PI3K pathway and corresponding biochemical activity, and how this relationship may be affected by crosstalk with other pathways that converge on similar transcriptional outputs.

Finally, based on the presented analyses, it will be of interest to evaluate the predictive power of a combined assessment of *PIK3CA* genotype and phenotypic PI3K/stemness scores in patient stratification for clinical trials with PI3K pathway inhibitors and, given the well-established implication of PI3K signaling in therapeutic response and resistance, with other cancer therapies.

## Materials and methods

### METABRIC and TCGA data access and pre-processing

Normalised microarray-based gene expression for METABRIC breast tumor samples were obtained from Curtis et al. [58], and clinical data from Rueda et al. [59]. The relevant METABRIC mutation data were downloaded from cBioPortal in January (mutation-only) and March (mutation and copy number) 2020 [60]. TCGA breast invasive carcinoma (BRCA) RNAseq, mutational and clinical data were retrieved from the GDC server (legacy database) using the *TCGAbiolinks* package [61], with additional mutation data retrieved from cBioPortal in January 2021 (for exact details, see the OSF-deposited RNotebooks). The *TCGAbiolinks* package was also used for subsequent quantile filtering (quantile value = 0.4; chosen empirically based on the observed count distributions) of lowly-expressed genes and removal of tumor samples with low purity (cpe = 0.6). The resulting raw RSEM counts were normalized with the TMM method [62] and log2-transformed using the voom() function in the *limma* package prior to downstream use in GSVA computations.

To analyze the relationships between *PIK3CA* genotype and PI3K/stemness scores, *PIK3CA* mutant datasets were subset for focus on hotspot *PIK3CA* variants only (C420R, E542K, E545K, H1047L, H1047R), excluding samples containing both a hotspot and a non-hotspot variant. The classification of hotspot vs non-hotspot variants was based on known clinical significance and frequency in patients with overgrowth caused by a single activating *PIK3CA* mutations [41]. Mutation data underwent manual checks to exclude samples with ambiguous genotype calls as well as all silent mutations.

We obtained information regarding allele amplification/gain in METABRIC breast tumors from cBioPortal and in TCGA breast tumors from our previous copy number analyses [8]. In

both cases, such information relies on a well-established computational deconvolution method known as ASCAT (allele-specific copy number analysis of tumors), which seeks to assign accurate allelic copy number to individual genomic regions while also estimating and adjusting for both tumor ploidy and normal cell admixture [63].

It remains difficult to ascertain that two or more mutations are present in the same cell as opposed to different cells. We note, however, that we have focused exclusively on hotspot *PIK3CA* variants for 'allele dosage' analyses, and such variants have been estimated to be clonal in breast cancer [64]. This strengthens the notion that our analyses are unlikely to be confounded by tumor mosaicism for different *PIK3CA* mutations.

## Calculation of transcription-based signature scores

The "HALLMARK_PI3K_AKT_MTOR_SIGNALING" and PluriNet gene sets were retrieved from The Molecular Signature Database (MSigDB) using the *mSigDBr* package [65]. Note that the "HALLMARK_PI3K_AKT_MTOR_SIGNALING" gene set also includes mTORC1-dependent gene expression changes, in contrast to other studies which have sought to separate AKT- and mTORC1-driven gene expression changes [45,46]. Categorization of scores into "low", "intermediate" and "high" was based on the 0.25 quantile, the interquartile range, and the 0.75 quantile, respectively. The stemness signature used by Miranda *et al.* [23] was retrieved from the accompanying supplementary material. The "PI3K_Jin_1" and "PI3K_Jin_2" gene signatures were obtained from Ref. [31]. Individual scores for each of these signatures were computed with the *GSVA* package, using the default Gaussian kernel and ESdiff enrichment values as output [30].

The PROGENy package was used to obtain a PI3K score according to a linear model based on pathway-responsive genes as described in Ref. [24].

The TCGAnalyze_Stemness() function in *TCGAbiolinks* was used to calculate a stemness score according to the machine learning model-based mRNAsi signature reported by Malta et al. [22].

## MCF10A breast epithelial cell culture

The generation of polyclonal non-transformed, immortalized breast epithelial MCF10A cells stably engineered to overexpress either empty vector (EV) or bovine hemagglutinin-tagged $PIK3CA^{H1047R}$ (retroviral vector: pJP1520-HA-PIK3CA(H1047R)) was described previously (see Supplementary Material of Ref. [32]). Cells were maintained in DMEM/F12 supplemented with the components indicated in **Table 1** and subcultured with Trypsin-EDTA (Fisher Scientific #MT-25-053-CI) at subconfluence. Cells were subcultured for two weeks post-thawing to ensure adaptation and used for experiments at passage 6–7. All treatments were performed according to a backwards design for simultaneous collection of all samples within a replicate run. Briefly, cells were seeded at 1500 cells/cm$^2$ (14000 cells per 6-well) on day 1, followed by

**Table 1. Culture medium composition for MCF10A breast epithelial cells.**

| Reagent | Vendor | Catalogue no. | Final concentration |
|---|---|---|---|
| Horse Serum | Gemini Bio | 100501 | 5% (v/v) |
| Gibco Recombinant AOF Insulin | Life Technologies | A11382II | 10 µg/ml |
| Hydrocortisone | Sigma | H4001 | 0.5 mg/ml |
| Recombinant hEGF | R&D Systems | 236-EG-01MAF-100-15 | 20 ng/ml final |
| Cholera toxin | List Biological Lab | 100B | 100 ng/ml final |
| DMEM/Ham's F12, with L-glutamine, phenol red, and sodium pyruvate | Wisent Bioproducts | 319-075-CL | 500 ml |

start of the 120 h DMSO or BYL719 (500 nM; Active Biochem #A-1214) on day 2, including full medium replacement for all cultures. Media replacement -/+ treatments was repeated on day 3 and 5. On day 5, 48 h treatments with DMSO or BYL719 were also initiated. On day 7, cells were washed once with 2 ml ice-cold PBS, followed by snap-freezing on dry ice and storage at -80°C until further processing. All media replacements and the final collection were performed at the same time of day to minimize biological noise.

## Western blotting

Cells were lysed in RIPA buffer (150 mM Tris-HCl, 150 mM NaCl, 0.5% (w/v) sodium deoxycholate, 1% (v/v) NP-40, pH 7.5) containing 0.1% (w/v) sodium dodecyl sulfate, 1 mM sodium pyrophosphate, 20 mM sodium fluoride, 50 nM calyculin, and 0.5% (v/v) protease inhibitor cocktail (Sigma-Aldrich) for 15 min. Cell extracts were precleared by centrifugation at 14,000 rpm for 10 minutes at 4°C. The Bio-Rad DC protein assay was used to assess protein concentration, and sample concentration was normalized using SDS sample buffer (62.5mM Tris pH 6.8, 2% SDS, 10% Glycerol, Bromophenol Blue) supplemented with 5% 2-mercaptoethanol (Sigma Aldrich #M3148-100ML) immediately before use. Lysates were resolved on acrylamide gels by SDS-polyacrylamide gel electrophoresis with PageRuler Plus pre-stained protein ladder (Fischer Scientific # PI26619) to approximate the size of separated proteins, then electrophoretically transferred to nitrocellulose membrane (BioRad) at 100 V for 90 min. Membranes were blocked in 5% (w/v) bovine serum albumin (Boston Bioproducts #P-753) in tris-buffered saline (TBS) for 1 h then incubated with specific primary antibodies diluted in 5% (w/v) bovine serum albumin in TBS-T (TBS with 0.05% Tween-20) at 4°C overnight, shaking. The next day, membranes were washed with TBS-T then incubated for 1 h at room temperature with fluorophore-conjugated secondary antibodies (LI-COR Biosciences). Details for all antibodies are provided in **Table 2**. The membrane was washed again with TBS-T then imaged with a LI-COR Odyssey CLx Imaging System (LI-COR Biosciences). Subsequent quantifications were performed in FIJI/ImageJ, by drawing a rectangle of the same size around each band of interest, as well as above it for background subtraction. The mean grey value was recorded and subtracted from 256, followed by subtraction of the background signal from the corresponding band signal. All targets were normalized to a corresponding total protein as indicated in the figure, in addition to normalization to the EV_DMSO condition within each experimental replicate and time point. All raw blot images and quantifications are deposited on the Open Science Framework and can be accessed via the following link: https://osf.io/dexgq/.

**Table 2. Primary and secondary antibodies used for Western blotting.** CST, Cell Signaling Technology. mAb, monoclonal antibody.

| Primary antibody (clone if mAb) | Vendor | Catalogue # | Lot # | Species | Size (kDa) | Dilution |
|---|---|---|---|---|---|---|
| p110α (C73F8) | CST | 4249 | 7 | rabbit | 110 | 1:1000 |
| pAKT S473 (D9E) | CST | 4060 | 24 | rabbit | 60 | 1:1000 |
| AKT (C67E7) | CST | 4691 | 20 | rabbit | 60 | 1:1000 |
| pPRAS40 T246 (C77D7) | CST | 2997 | 12 | rabbit | 40 | 1:1000 |
| PRAS40 (D23C7) | CST | 2691 | 11 | rabbit | 40 | 1:1000 |
| pS6 S240/S244 (D68F8) | CST | 5364 | 7 | rabbit | 32 | 1:1000 |
| S6 (5G10) | CST | 2217 | 9 | rabbit | 32 | 1:1000 |
| Vinculin (E1E9V) | CST | 13901 | 6 | rabbit | 124 | 1:1000 |
| **Secondary antibody** | | | | | | |
| IRDye 800CW Goat anti-Rabbit IgG (H + L) | LI-COR | 926–32211 | D00825-14 | goat | | 1:20,000 |

## MCF10A RNA sequencing and data analyses

Snap-frozen cells were thawed on ice and processed for RNA extraction with Takara's Nucleospin RNA Plus (#740984.50) according to the manufacturer's instructions. Samples were submitted to Novogene for quality control (Agilent 2100 analysis), mRNA library preparation (unstranded) and final paired-end sequencing on an Illumina NovaSeqS4 lane.

Raw read processing was performed with the Nextflow (version 20.07.1) nf-core RNAseq pipeline (version 1.4.2) [67], with Spliced Transcripts Aligment to a Reference (STAR) [68] for read alignment to the human genome (Homo_sapiens.GRCh38.96.gtf) and featureCounts [69] for counting of mapped reads (multimapped reads were discarded). Subsequent data processing was performed in R according to the limma-voom method [70]. Briefly, raw counts were converted to counts per million (cpm) using the cpm() function in the *edgeR* package [71], followed filtering of lowly expressed genes using the *TCGAbiolinks* package with quantile values 0.80 (chosen empirically based on the observed count distribution); results using more stringent and more lenient filtering options are also included in S1 Fig. Next, read count normalization was performed with the trimmed mean of M (TMM) method [62]. One sample was removed due to a low total read count and outlier behavior upon unsupervised dimensionality reduction (principal component analysis with the *PCAtools* package). The mean-variance relationship was modelled with voom(), followed by linear modelling and computation of moderated t-statistics using the lmFit() and eBayes() functions in the *limma* package [70]. Experimental replicate was included as a batch effect term in the model. The associated p-values for assessment of differential gene expression were adjusted for multiple comparisons with the Benjamini-Hochberg method at false-discovery rate (FDR) = 0.05 [72].

## Gene set enrichment analysis (GSEA)

GSEA on MCF10A and iPSC transcriptomic data was performed in R, on the list of all genes ranked according to the *t* statistic for each comparison of interest; the choice of *t* statistic ensures that the gene ranking considers signal (fold change) as well as noise. The iPSC gene lists were obtained from Ref. [43]. Normalized enrichment values and associated p-values were calculated using *fgsea* with 100,000 permutation (nperm = 100000) [73]. The normalized enrichment score computed by *fgsea* corresponds to the enrichment score normalized to mean enrichment of random samples, using the same gene set size. Note that GSEA was used instead of GSVA for these analyses as the latter is recommended for use with relatively large sample sizes (n > 30) and experimental designs beyond conventional case-control set-ups [30]. Whereas GSEA based on *t* value rankings evaluates the concerted differential expression of a set of genes relative to all other genes within a given phenotypic comparison ("supervised"), the GSVA method evaluates the absolute expression of a set of genes relative to all other genes and does not require *a priori* phenotypic comparisons to be specified. Both methods are classified as competitive since the enrichment score in each case is calculated as a function of gene expression inside and outside a given gene set. It is important to note that statistical significance and absolute scores for individual GSEA enrichment scores can be highly dependent on prior gene filtering choices as shown in **S1 Fig** and covered in detail in Ref. [74].

## Statistical analyses

Linear models were used to assess the significance of the relationship between stemness and PI3K scores in both METABRIC and TCGA breast cancer cohorts. One-way ANOVA followed by Tukey's Honest Significant Differences (HSD) method was used to perform pairwise significance testing with multiple comparison adjustments (adjusted p-value < 0.05) when

evaluating grade- and cancer subtype-specific differences in PI3K/stemness scores across the METABRIC cohort; similar analyses were not performed with the TCGA breast cancer data due to smaller sample size and incomplete grading information. ANOVA with Tukey's HSD was also used to evaluate the significance of the relationships between *PIK3CA* genotype and PI3K/stemness scores across both cohorts. For linear models as well as ANOVAs, the residuals were examined to confirm that model assumptions were met. The only assumption that was violated was that of normality; however, given the large sample size, this violation is expected to have a minimal impact on model validity [75].

Differences in categorial PI3K/stemness score ("low", "intermediate", "high") distributions across tumor subtypes and/or genotypes were assessed using a Chi-squared goodness-of-fit test. The relationship between PI3K/scores and survival was assessed using a non-parametric log-rank test.

Pairwise correlation analyses and hierarchical clustering of signature scores were performed using Spearman's rank correlation and the Ward.D2 method (available through R package *corrplot*; *https://github.com/taiyun/corrplot*). The associated p-values were adjusted for multiple comparisons using the Bonferroni method (family-wise error rate < 0.05).

Statistical analyses pertaining to MCF10A RNA sequencing and GSEA are described in the relevant sections above.

## R packages information

As indicated in the accompanying scripts, all relevant packages were sourced either from CRAN or Bioconductor (via BiocManager [76]). Figures were produced using the *ggplot2* package [77].

## Supporting information

**S1 Fig. The effect of background gene filtering on the GSEA output.** Each plot corresponds to replicate analyses of the MCF10A transcriptomic data in Fig 2 (main manuscript), following different filtering thresholds for absolute gene expression. The total number of ranked genes and their overlap with the tested signatures are shown above each analysis. The p-values correspond to each enrichment's significance following 100,000 permutations of the gene ranks; * $p \leq 0.05$, ** $p \leq 0.01$, *** $p \leq 0.001$; FDR = 0.05.
(TIFF)

**S2 Fig. (A)** PI3K signaling score distribution in TCGA breast tumors stratified according to ER status. Survival analysis in estrogen receptor (ER)-negative breast cancer patients, as a function of PI3K signaling **(B)** or stemness **(C)** score. **(D)** Pan-breast cancer patient survival in TCGA, as a function of PI3K activity score. **(E)** ER-negative breast cancer patient (METABRIC) survival as a function of binary *PIK3CA* genotype. The sample size for each panel and subgroup is indicated, and p-values were calculated using a log-rank test; where shown, the 95% confidence intervals are indicated by shading. **(F)** UpSet plot showing intersection set sizes across the specified gene set combinations.
(TIFF)

**S1 Table. mSigDB "HALLMARK_PI3K_AKT_MTOR_SIGNALING" gene list.**
(CSV)

**S2 Table. mSigDB "MUELLER_PLURINET" gene list.**
(CSV)

## Acknowledgments

We are grateful to Dr Neil Vasan (Columbia University, New York) and members of the Saez-Rodriguez group in Heidelberg for excellent feedback on the manuscript. We would also like to thank the cancer community behind the TCGA/METABRIC datasets, as well as data scientists developing the above-mentioned analysis tools, for making them publicly available and thus enabling the completion of this study.

## Author Contributions

**Conceptualization:** Ralitsa R. Madsen.

**Data curation:** Ralitsa R. Madsen, Emily C. Erickson.

**Formal analysis:** Ralitsa R. Madsen.

**Funding acquisition:** Ralitsa R. Madsen, Emily C. Erickson, Carlos Caldas, Alex Toker, Robert K. Semple, Bart Vanhaesebroeck.

**Investigation:** Ralitsa R. Madsen, Emily C. Erickson.

**Methodology:** Ralitsa R. Madsen.

**Project administration:** Ralitsa R. Madsen.

**Resources:** Carlos Caldas.

**Supervision:** Oscar M. Rueda, Xavier Robin, Alex Toker, Robert K. Semple, Bart Vanhaesebroeck.

**Visualization:** Ralitsa R. Madsen.

**Writing – original draft:** Ralitsa R. Madsen.

**Writing – review & editing:** Ralitsa R. Madsen, Emily C. Erickson, Oscar M. Rueda, Xavier Robin, Alex Toker, Robert K. Semple, Bart Vanhaesebroeck.

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
