## [Decision Letter · Decision Letter 0]

9 Aug 2021

Dear Dr Madsen,

Thank you very much for submitting your Research Article entitled 'Close correlation between transcriptomically-inferred stemness and PI3K/AKT/mTOR signalling scores in breast cancer, and a counterintuitive relationship with PIK3CA genotype' to PLOS Genetics.

The manuscript was fully evaluated at the editorial level and by independent peer reviewers. The reviewers appreciated the attention to an important problem, but raised some substantial concerns about the current manuscript. Based on the reviews, we will not be able to accept this version of the manuscript, but we would be willing to review a much-revised version. We cannot, of course, promise publication at that time.

Should you decide to revise the manuscript for further consideration here, your revisions should address the specific points made by each reviewer. [Please note that in a revised manuscript it will be essential to adequately address the remaining concerns of reviewer #1].  We will also require a detailed list of your responses to the review comments and a description of the changes you have made in the manuscript.

If you decide to revise the manuscript for further consideration at PLOS Genetics, please aim to resubmit within the next 60 days, unless it will take extra time to address the concerns of the reviewers, in which case we would appreciate an expected resubmission date by email to plosgenetics@plos.org.

[LINK]

We are sorry that we cannot be more positive about your manuscript at this stage. Please do not hesitate to contact us if you have any concerns or questions.

Yours sincerely,

Peter McKinnon

Section Editor: Cancer Genetics

PLOS Genetics

David Kwiatkowski

Section Editor: Cancer Genetics

PLOS Genetics

Reviewer's Responses to Questions

**Comments to the Authors:**

Reviewer #1: The revised manuscript from Madsen et al does honestly quite little to address what I still believe to be reasonable concerns about the paper. I do agree with the other two reviewers that the conceptual point, which I understand to be that higher dimensional markers of PI3K activity (“scores”) are likely to be more clinically useful and indicative of response to anti-PI3K drugs, in comparison to the simple single genetic mutation marker, is a useful and important concept. This paper does have analyses that support this idea. Yet, I still contend that an important control is to ask the question of how many other such “scores” show the types of trends that are observed, but by chance? This is not simply looking for another alternative PI3K score as was done by the authors. Perhaps, this kind of rigor is beyond scope in the opinions of the editors and other reviewers, but to me it does seem warranted to understand the fidelity of the finding, particularly given the absence of experimental follow up. Further, I apologize for the lack of clarity in the type of experimental evidence I thought might be needed to make the paper more sufficiently convincing. To me, cell line and cell culture experiments would be perfectly adequate…I was not stating new animal / tissue / human data was needed for this paper. However, I still find a major gap in the paper to be that no phenotypic or signaling markers of PI3K/mTOR/AKT activity (e.g. S473 phosphorylation on AKT) are measured to show the scores have validated meaning, and/or whether cell lines with different mutations in PI3K and scores have differential responses to PI3K inhibitors. Those should be relatively straightforward cell culture experiments. The only cell line data I found in the paper was for stem cell lines that were homo or heterozygous for PI3K H1047R (Fig. 4D), showing they have different PI3K and stemness scores. Thus, while overall it seems that I am in agreement more than not with the other two reviewers, I may still differ in my opinions as to additional analyses that would help make this paper convincing enough to have wide impact. Yet, I am OK if the others disagree with that assessment and find the revised manuscript to be satisfactory.

Reviewer #2: I am happy to recommend acceptance. The authors have responded carefully and thoughtfully to the 3 reviewers' critiques, and have made the appropriate changes to the manuscript. I have no concerns to communicate.

**Have all data underlying the figures and results presented in the manuscript been provided?**

Reviewer #1: Yes

Reviewer #2: Yes

PLOS authors have the option to publish the peer review history of their article (what does this mean?). If published, this will include your full peer review and any attached files.

Reviewer #1: No

Reviewer #2: No

---

## [Decision Letter · Decision Letter 1]

13 Oct 2021

Dear Dr Madsen,

We are pleased to inform you that your manuscript entitled "Positive correlation between transcriptomic stemness and PI3K/AKT/mTOR signaling scores in breast cancer, and a counterintuitive relationship with PIK3CA genotype" has been editorially accepted for publication in PLOS Genetics. Congratulations!

Yours sincerely,

Peter McKinnon

Section Editor: Cancer Genetics

PLOS Genetics

Comments from the reviewers (if applicable):

Reviewer's Responses to Questions

**Comments to the Authors:**

Reviewer #1: It seems the authors revision shows a correlation of PI-3K pathway activation in cell culture models with established GSEA sets (Figure 2). Much of the author's claims in the paper revolve around PI-3K activity score based on GVSA. The distinction between the two is hard to decipher, but I am fine with publication.

**Have all data underlying the figures and results presented in the manuscript been provided?**

Reviewer #1: Yes

PLOS authors have the option to publish the peer review history of their article (what does this mean?). If published, this will include your full peer review and any attached files.

Reviewer #1: No

**Data Deposition**

http://datadryad.org/submit?journalID=pgenetics&manu=PGENETICS-D-21-00915R1

**Press Queries**

---

## [Editor Report · Acceptance letter]

21 Oct 2021

PGENETICS-D-21-00915R1 

Positive correlation between transcriptomic stemness and PI3K/AKT/mTOR signaling scores in breast cancer, and a counterintuitive relationship with PIK3CA genotype 

Dear Dr Madsen, 

We are pleased to inform you that your manuscript entitled "Positive correlation between transcriptomic stemness and PI3K/AKT/mTOR signaling scores in breast cancer, and a counterintuitive relationship with PIK3CA genotype" has been formally accepted for publication in PLOS Genetics! Your manuscript is now with our production department and you will be notified of the publication date in due course.

With kind regards,

Livia Horvath

PLOS Genetics

On behalf of:
